# Efficient Transformer-Based Compressed Video Modeling via Informative Patch Selection

**DOI:** 10.3390/s23010244

**Published:** 2022-12-26

**Authors:** Tomoyuki Suzuki, Yoshimitsu Aoki

**Affiliations:** Department of Electronics and Electrical Engineering, Faculty of Science and Technology, Keio University, 3-14-1, Hiyoshi, Kohoku-ku, Yokohama 223-8522, Kanagawa, Japan

**Keywords:** video recognition, action recognition, transformer, compressed video

## Abstract

Recently, Transformer-based video recognition models have achieved state-of-the-art results on major video recognition benchmarks. However, their high inference cost significantly limits research speed and practical use. In video compression, methods considering small motions and residuals that are less informative and assigning short code lengths to them (e.g., MPEG4) have successfully reduced the redundancy of videos. Inspired by this idea, we propose Informative Patch Selection (IPS), which efficiently reduces the inference cost by excluding redundant patches from the input of the Transformer-based video model. The redundancy of each patch is calculated from motions and residuals obtained while decoding a compressed video. The proposed method is simple and effective in that it can dynamically reduce the inference cost depending on the input without any policy model or additional loss term. Extensive experiments on action recognition demonstrated that our method could significantly improve the trade-off between the accuracy and inference cost of the Transformer-based video model. Although the method does not require any policy model or additional loss term, its performance approaches that of existing methods that do require them.

## 1. Introduction

The development of neural-network-based video models has improved the accuracy of video recognition. Recently, video models based on Transformers [1] have been proposed [2,3,4,5]. Transformer first revolutionalized the natural language processing field [6], and in image recognition, methods based on it have shown high levels of effectiveness on multiple tasks in a wide range of domains from natural images [7,8] to medical images [9,10], industrial material images [11] and PoISAR images [12]. In addition, in video recognition, Transformer-based video models are comparable or superior to Convolutional Neural Network (CNN)-based models [13,14,15,16,17,18,19,20,21], which have been predominant in video recognition for a decade. Furthermore, Transformer-based models are known to scale their performance with increasing amounts of training data compared to CNN-based models [7,22]; therefore, they have promising potential given the recent trend of rapidly scaling training and pre-training data [23,24,25,26,27]. On the other hand, the high inference cost of Transformer-based video models significantly limits the efficiency of research and practical use. Therefore, it is necessary to improve its trade-off between accuracy and inference cost.

A great deal of information is shared in a spatiotemporal neighborhood of videos, making them highly redundant. This is because videos are represented as appearance and motion, and motions follow a distribution that is biased toward the spatial neighborhoods (i.e., zero-centered). Therefore, when a frame of a video is represented by motions and residuals from the previous frame, the areas with small motions and residuals are dominant. In video compression, the method considering such areas include little information content, and assigning short code lengths to them (e.g., MPEG4 [28]) has been successful and is widely used. However, most conventional video recognition models do not take this redundancy into account and input them as full-frame sequences. In video recognition, as in video compression, is it possible to dramatically improve efficiency by taking redundancy into account?

Our main contribution is the proposal of **I**nformative **P**atch **S**election (IPS), which efficiently reduces the inference cost of Transformer-based video models by excluding patches with little information content from the input. More precisely, IPS models the distribution of motions and residuals in a patch of a P-frame (Predicted frame) from an MPEG4-encoded video. Then, patches with a large probability are considered to have a small amount of information and are excluded from the input. IPS is simple and effective in that it can *dynamically* reduce inference cost depending on the input *without requiring any policy model and complicated training*, unlike the previous works [29,30,31,32,33,34,35,36,37,38,39,40]. Modeling MPEG4-encoded video has long been conducted for network traffic prediction [41,42,43,44], and recently, neural-network-based video recognition models directly taking MPEG4-encoded video as input were proposed [5,45]. However, to the best of our knowledge, our work is the first attempt to utilize MPEG4-encoded video information for *optimizing the dynamic inference costs of video recognition*.

Since the inference cost of a vanilla Transformer-block [1] increases with the number of input elements (frame patches in case of video), reducing the number of input patches can result in the reduction in inference cost. However, that this can be applied to recent Transformer-based video models like TimeSformer [2] is nontrivial because they often contain a variant of Transformer-block (e.g., dividing spatiotemporal attention into space and time) to improve efficiency. Therefore, in preliminary experiments, we demonstrated that reducing the number of input patches reduces the inference cost in Transformer-based video models with the major variants of Transformer-block.

In addition, as in [5], we experimentally confirmed the effectiveness of inputting P-frames as motions and residuals into the model for action recognition. When a P-frame is represented as motion and residuals, and some patches are not observed due to patch selection, the problem is that it is impossible to identify corresponding regions in an I-frame (Intra-coded frame) of the observed patches in a P-frame. To solve this problem, we applied P-frame accumulation [45], which makes all motions and residuals stem from the previous I-frame by backtracing in the time direction.

We used multiple action recognition datasets to evaluate the effectiveness of IPS. Through extensive ablation studies, we explored the optimal design of IPS. We also demonstrated the effectiveness of IPS by comparing its trade-offs between accuracy and inference cost against those of multiple baseline models. Finally, IPS enables the achievement of a 71.8% reduction in inference cost with only a 0.89% drop in accuracy on HMDB-51 dataset, a 69.2% reduction in inference cost with only a 1.93% drop in accuracy on SSv2, and an 80.7% reduction in inference cost with only a 1.01% drop in accuracy on Mini-Kinetics. The TimeSformer with IPS is significantly more efficient than conventional Transformer-based models. Moreover, even though IPS does not require any policy model or additional loss, it achieves performance approaching that of existing efficient CNN-based methods that do require them.

The rest of this paper is organized as follows. Section 2 reviews related work focusing on efficient video recognition. Our methodology is presented in Section 3. The experimental results are presented and discussed in Section 4. Finally, in Section 5, we summarize the proposed method and discuss future works.

## 2. Related Works

This section offers a review of related research related to the goal of this study, that is, improving the efficiency of video recognition—and more precisely, optimizing the trade-off between accuracy and inference cost.

Optimizing the model architecture is the most common approach to improving video recognition efficiency, though our method is not in this line of approach. Three-dimensional CNNs [13,18,19], which are neural networks equipped with 3D convolution, represent the most basic video recognition models. However, these are computationally expensive, and attempts to optimize their operations and architectures to reduce inference cost have been made for a decade. These can be either manually designed, such as by dividing space-time convolutions into space and time [20,21], or learning-based, such as neural architecture searches in CNN-based models [46]. In the lower-computation domain, video recognition models based on 2D CNNs come into play. TSN [14], TRN [15] and ECO [47] are simple baselines that aggregate frame-level 2D CNN features to output a video-level label. TSMNet [16,17] is a 2D CNN equipped with temporal shift modules, which simply shifts a part of channels along the temporal dimension. MSNet [48] is a 2D CNN equipped with motion squeeze modules, which extracts flow features between adjacent frames, in addition to temporal shift modules. Recently, video models based on Transformer [1], which have shown high levels of effectiveness in language processing [6] and image recognition [7,8], have been proposed. For these models, methods of improving the efficiency of attention computation have been proposed. In TimeSformer [2], Divided Space-Time attention, in which temporal and spatial attention are applied alternately, is used. ViViT [3] uses Factorized Encoder In ViViT [3], which is a Transformer-based video model almost identical to TimeSformer, Factorized Encoder is used. It calculates the frame-level embedding vectors by first only applying attention to the spatial direction, and then these vectors are related only by temporal attention. In the line of research seeking to improve the efficiency of attention calculation, the most recent work is the Video Swin Transformer [49]. It equips local attentions that can be efficiently implemented with shifted windows. Although these approaches are meant to improve the efficiency of *attention matrix calculations*, we show that *element-wise calculations* also account for much of the inference cost. We then reduce the computational cost, including element-wise calculation, by reducing the number of input patches. In addition, the methods described above are static to the input. In contrast, our approach dynamically reduces inference cost depending on the input.

Another approach to improving video recognition efficiency is input-size optimization, and our method falls under this kind of this approach. These approaches usually focus on dynamic optimizations of input-size using additional policy models or modules. For example, they select salient frames/clips as input [29,30,31,32]; adjust the resolutions [33], precision of network quantization [34] and channel fusion [35]; predict crop regions to input [36,37,38]; or determine whether to exit early in the time direction [39] or to use the heavy teacher knowledge [40]. Our method, on the other had, selects spatiotemporal patches to input. These methods are somewhat complicated in that they require the interdependent learning of the recognition model or additional loss functions. The proposed method is simpler to learn and therefore more practical than conventional methods because, unlike those methods, it does not require any policy model or additional loss. As a method specializing in Transformer-based models, VidTr [4], which is based on a stack of decomposed spatiotemporal attentions similar to TimeSformer and ViViT, selects frames to process based on the standard deviation of the attention weights. In contrast, since our method can select inputs based on finer units, i.e., *patches*, it is expected to reduce inference cost more efficiently.

In addition to the approaches described above, CoViAR [45] uses motions and residuals obtained in the process of decoding compressed video as input. The overall inference cost reduction is achieved by partially omitting the decoding process and optimizing the model size for each modality. Expanding this to a Transformer-based video model, MM-ViT [5] takes multi-modal input, including audio and frames represented by RGB+motions+residuals from compressed videos. Like MM-ViT, the proposed method utilizes information obtained during the process of decoding compressed video. However, our idea of using motions and residuals for input-size optimization is the first of its kind.

## 3. Methodology

Our goal is to develop an efficient (i.e., low inference cost) method that takes a compressed video as input and outputs some label (e.g., action class) associated with it. We adopt TimeSformer [2], one of the basic Transformer-based video models, for a recognition model. We efficiently reduce the inference cost of the Transformer-based video model via IPS, which excludes patches with little information content based on motions and residuals obtained in the process of decoding the video. Figure 1 shows the overview of our modeling. First, I-frames and P-frames in a compressed video are decoded into RGB values and motions+residuals, respectively. Then, while all the patches are inputted into a Transformer-based video model for the I-frames, for P-frames, only the patches selected by IPS are inputted.

In this study, we assume that the videos to be inferred are stored in a compressed state. Most videos in the real world are stored in a compressed state because they occupy more storage when stored as frame images. Therefore, it is reasonable to make this assumption.

In the rest of this section, we present the details of our methodology. First, Section 3.1 gives a brief illustration of motions and residuals in MPEG4 videos. Second, we explain our overall modeling, including how IPS is incorporated into the overall model in Section 3.2. Then, in Section 3.3, we explain the impact of reducing input patches on the inference cost of TimeSformer, which results in reducing the input patches needed by IPS. Finally, in Section 3.4, we give a detailed explanation of IPS.

### 3.1. Preliminary: Motions and Residuals

Like [45], we assume the use of MPEG4 [28], a widely used video compression method. MPEG4 divides a video into group of pictures (GoP), where each groups consists of 12 consecutive frames. The first frame of a GoP is an I-frame represented by RGB luminance values $v^{\left(t\right)}\in R^{H\times W\times 3}$, and the rest is represented as a P-frame consisting of the motions from the previous frame $m^{\left(t\right)}\in R^{H\times W\times 2}$ and the residual $r^{\left(t\right)}\in R^{H\times W\times 3}$ , where t∈{1,...,12} is a number indicating the temporal index of frames in GoP, where t=1 is an I-frame and the rest are P-frames. Because a motion is calculated in units of equal-sized patches, the spatial resolution is smaller than the frame size. However, for simplicity of explanation, it is represented as if it were resized to the same size as the frame. P-frames can be reconstructed as RGB values by adding the residuals to the corresponding RGB values of the previous frame obtained from the motion:(1)$$\begin{array}{c} v_{\left(h,w\right)}^{\left(t\right)}=v_{\left(h,w\right)-m_{\left(h,w\right)}^{\left(t\right)}}^{\left(t-1\right)}+r_{\left(h,w\right)}^{\left(t\right)}, \end{array}$$
where $v_{\left(h,w\right)}^{\left(t\right)}$ and $m_{\left(h,w\right)}^{\left(t\right)}$ and $r_{\left(h,w\right)}^{\left(t\right)}$ represent the RGB values, motions and residuals of the (h,w) spatial coordinate in *t*-th frame, respectively. Because I-frames retain RGB values, all P-frames can be restored to RGB values by recursively applying the equation presented above. Due to the nature of video, the motions and residuals from the previous frame tend to have distributions biased toward zero; therefore, the compression effect via entropy encoding is greater than that of the RGB values.

### 3.2. Overview of Modeling

In this section, we explain our entire modeling method. First, we give an illustration of the original TimeSformer. Second, the method of inputting P-frames as motions and residuals into TimeSformer is described. Finally, we explain the method to incorporate IPS into TimeSformer.

#### 3.2.1. Original TimeSformer

We adopt TimeSformer [2] as the basic model. The input to TimeSformer is a sequence of frame images $x\in R^{N\times D}$ divided into patches of size P×P, where $N=THW/P^{2}$ is the number of patches, $D=CP^{2}$ is the number of dimensions representing the patch and *C* is the number of original channels in the frame image.

These are linearly transformed and added to the corresponding trainable position embedding to obtain patch embedding $z\in R^{N\times D}$:(2)$$\begin{array}{ccc} z_{\left(i\right)} & = & W\cdot x_{\left(i\right)}+p_{\left(i\right)}, \end{array}$$
where $W\in R^{D\times D}$ is a trainable weight matrix and $x_{\left(i\right)}$ and $p_{\left(i\right)}$ are the patch and position embedding corresponding to *i*-th.

Next, we concatenate patch embeddings z and a trainable classification token $c\in R^{1\times D}$ on the first axis. In this paper, we call concatenated patches and classification tokens an “element“ and refer to this first axis as the element-axis. We input them into encoder $E_{L}$ equipped with *L* self-attention Transformer blocks and obtain video feature $z^{L}$:(3)$$\begin{array}{ccc} z^{L} & = & E_{L}\left(\mathrm{ElementConcat}\left(c,z\right)\right)\in R^{(N+1)\times D}, \end{array}$$
where ElementConcat is the concatenation on the element-axis.

Finally, the output per video is calculated using the first element $z_{\left(0\right)}^{L}$ , which corresponds to the classification token. For example, in the action recognition task (i.e., classification into action classes), the output is a class probability vector computed using a classifier consisting of a layer normalization, a multiple layer perceptron $\phi_{\mathrm{MLP}}$ and a softmax function:(4)$$\begin{array}{ccc} y & = & \mathrm{Softmax}\left(\phi_{\mathrm{MLP}}\left(\mathrm{LayerNorm}\left(z_{\left(0\right)}^{L}\right)\right)\right)\in R^{D_{\mathrm{class}}}, \end{array}$$
where $D_{\mathrm{class}}$ is the number of classes. The computational cost of encoder $E_{L}$ accounts for a large portion of TimeSformer’s overall inference cost and is subject to a reduction via IPS.

#### 3.2.2. TimeSformer with P-Frames as Motions and Residuals

TimeSformer assumes inputs are frame images represented by RGB values, but P-frames can also be input as motions and residuals instead of RGB images. Note that a method for directly inputting motions and residuals into a Transformer-based model already exists [5]; this is not our main contribution. In [5], the multimodal information, including audio and frames represented by RGB+motions+residuals from a compressed video, is input into MM-ViT equipped with complex intermodality attentions. We treated P-frames as motions and residuals in a much simpler way but still achieved a solid improvement in accuracy, as shown in the experiments (in Section 4.3.1). First, to keep the computational cost the same as that of RGB, we average residuals in the channel direction and concatenate them with the motions in the channel direction, representing P-frames as three-channel frame images. Moreover, the only modification to the model is the change in the weights of the linear projection to make patch embeddings depending on input frame types (i.e., I-frame or P-frame):(5)$$\begin{array}{c} z_{\left(j\right)}^{I}=W^{I}\cdot x_{\left(j\right)}^{I}+p_{\left(j\right)}^{I} \end{array}$$(6)$$\begin{array}{c} z_{\left(k\right)}^{P}=W^{P}\cdot x_{\left(k\right)}^{P}+p_{\left(k\right)}^{P} \end{array}$$(7)$$\begin{array}{c} z=\mathrm{ElementConcat}(z^{I},z^{P}), \end{array}$$
where $x^{I}\in R^{N^{I}\times D}$ and $x^{P}\in R^{N^{P}\times D}$ are patches of I-frames and P-frames; $N^{I}$ and $N^{P}$ are the number of patches belonging to I-frames and P-frames (i.e., $N=N^{I}+N^{P}$); $p^{I}\in R^{N^{I}\times D}$ and $p^{P}\in R^{N^{P}\times D}$ are trainable position embeddings for I-frames and P-frames; and $W^{I}\in R^{D\times D}$ and $W^{P}\in R^{D\times D}$ are the weights of the linear projection for I-frames and P-frames, respectively.

#### 3.2.3. TimeSformer with IPS

IPS reduces the inference cost by selecting only a part of the patch embedding and reducing the number of patches input into the encoder $E_{L}$. Let *S* be the set of patch indices selected by IPS; Equations (7) and (Equation 3) are modified as follows:(8)$$\begin{array}{ccc} \hat{z} & = & \mathrm{ElementConcat}\left(z^{I},{\left[z_{k}^{P}\right]}_{k\in S}\right)\in R^{(N^{I}+|S|)\times D} \end{array}$$(9)$$\begin{array}{ccc} z^{L} & = & E_{L}\left(\mathrm{ElementConcat}\left(c,\hat{z}\right)\right)\in R^{(N^{I}+|S|+1)\times D}, \end{array}$$
where $|S|<N^{P}$ and |S| varies adaptively with the input (see Section 3.4 for details).

### 3.3. Impact of Reducing Input Patches on Inference Cost of TimeSformer

Before describing IPS, we explain the degree to which inference cost can be reduced by reducing the number of patches input into TimeSformer. TimeSformer proposes several types of spatiotemporal attention [2]. We adopt Joint Space-Time and Divided Space-Time attentions, which can consider the spatiotemporal relationship of patches and achieve high recognition accuracy. Joint Space-Time attention computes the attention over the entire set of input patches, whereas Divided Space-Time alternately applies space and time attention, thereby improving computational efficiency but including almost twice the number of parameters as Joint Space-Time attention.

In addition, multiple forms of spatiotemporal attention have been proposed in ViViT [3], which, like TimeSformer, regards videos as a series of spatiotemporal patches to be modeled by Transformer blocks. From these forms of spatiotemporal attention, we employ Factorized Encoder for TimeSformer, which has been concluded to be the most efficient in terms of the trade-off between accuracy and inference cost, in addition to Joint Space-Time and Divided Space-Time attentions. Factorized Encoder first computes frame-by-frame embedding vectors using space attention and then relates them using time attention.

For the Transformer block with regular attention used in Joint Space-Time attention, all the internal processing is directly reduced by reducing the number of input patches. However, this is not the case when attention is applied to only one axis (target axis), such as the the time axis or space axis used in the Divided Space-Time and Factorized Encoder (we call this ”axis attention”). A PyTorch-style pseuducode for axis attention is shown in Algorithm 1. Axis attention improves efficiency by parallelizing the attention matrix calculation (”attention matrix calculation” in Algorithm 1) for the other axis (the nontarget axis). However, to realize this parallel computation on Graphics Processing Units (GPUs), which are widely used computational devices, the length of the target axis must be consistent over the nontarget axis. When a portion of the input patch is selected by IPS, the target axes’ lengths will not be aligned, forcing the use of padding tensors to align them for parallelization. As a result, the reduction in computational cost in the attention matrix calculation is significantly limited.

Nevertheless, this does not mean that it is impossible to reduce the inference cost of Divided Space-Time attention and Factorized Encoder by reducing the number of input patches. Element-wise calculation in a Transformer block (e.g., linear projections and a multi-layer perceptron), which can be directly reduced by reducing the number of input patches, accounts for the overall inference cost in the Transformer block, as well as the attention matrix calculation. As shown in Algorithm 1, we reduced the inference cost for the Transformer block with axis attention by skipping only the element-wise calculation of excluded patches.

Figure 2 shows the Giga FLoating-Operation-Points (GFLOPs) per video of the attention matrix calculations and element-wise calculations in TimeSformer for each resolution and type of attention when changing the percentage reduction in the number of patches for P-frames. Here, the number of blocks in the Transformer is set to 12, the input is 1 GoP (i.e., 1 I-frame and 11 P-frames) and the patch size is set to 16. The maximum number of patches increases with the resolution. Note that the computation costs of other operations within TimeSformer are negligibly small compared to the attention of matrix calculations and element-wise calculations.

Joint Space-Time attention reduces the cost of both the attention matrix calculation and element-wise calculation by reducing the number of input patches, resulting in a significant reduction in the overall inference cost. In Divided Space-Time attention and Factorized Encoder, the attention matrix calculation has already been made more efficient. However, element-wise calculation still accounts for much of the computational cost. We can see that reducing the number of patches can significantly reduce the overall inference cost by reducing element-wise calculations. Note that because in Divided Space-Time attention the spatiotemporal relationship is modeled with two Transformer blocks, one in the temporal direction and the other in the spatial direction, the element-wise calculation is almost twice as large as that of Joint Space-Time attention.
**Algorithm 1:** PyTorch-like pseudocode of a Transformer block with axis attention (spatial) and IPS. Lists of numbers in parentheses in comments indicate the shapes of tensors.# Inputs:# T: number of time indexes# S: number of space indexes# D: number of channels# z_prev: embeddings of elements from the previous Transformer block (T*S+1, D)# mask: mask of elements. False if selected by IPS else True (T*S) # initialize embedding (z) for spatial axis attentionz = z_prevz_cls = z[:1].unsqueeze(0).expand(T, -1, -1)# (T, 1 ,D)z_patch = z[1:].reshape(T, S, D)# (T, S ,D)z = concat(z_cls, z_patch), axis=1)# (T, S+1 ,D) # reshape mask and pad it for classification tokenmask = concat(zeros([T, 1], dtype=bool), mask.reshape(T, S), axis=1) # (T, S+1) # get selected (not masked) indicesind = where(not mask) # — element-wise calculation —# calculate query, key and value for only selected indicesq, k, v = zeros([T, S+1, D]), zeros([T, S+1, D]), zeros([T, S+1, D])q[ind], k[ind], v[ind] = fc_q(z[ind]), fc_k(z[ind]), fc_v(z[ind]) # — attention matrix calculation —# calculate attention matrix ”in parallel” along temporal-axis (axis=0)attn = (q @ k.transpose()) / scale# (T, S+1, S+1)mask = mask.unsqueeze(1).expand(-1, S+1, -1)# (T, S+1, S+1)attn = attn.masked_fill(mask=mask, value=float(’-inf’))attn = softmax(attn, axis=1)z = attn @ v# (T, S+1 ,D) # — element-wise calculation —# postprocessz = fc(z)z_cls = mean(z[:, :1], axis=1)# averaging along temporal-axisz = concat(z_cls, z[:, 1:])z += z_prevz = MLP(LayerNorm(z))z_prev = z

Attention matrix calculations, which increase computational complexity in proportion to the square of the number of elements, are frequently regarded as the largest bottleneck in cost-reduction efforts [2,3]. However, element-wise calculations also account for a large portion of the cost, especially when the number of patches is small. In addition to the engineering of the attention matrix calculation, the simple idea of reducing the number of input patches contributes significantly to reducing the inference cost. Note that although we use TimeSformer as a representative Transformer-based video model, we believe that reducing the number of patches also contributes to reducing the inference cost of other methods, which model spatiotemporal patches of a video with Transformers, such as ViViT [3], VidTr [4] and MM-ViT [5].

### 3.4. Informative Patch Selection (IPS)

In a video, a great deal of information is shared in spatiotemporal neighborhoods; that is, a certain pixel value is often easily predictable from its neighborhood. This is due to the fact that videos are represented as appearances and their motions, and motions follow a distribution that is biased toward the spatial neighborhoods (i.e., zero-centered). In widely used video compression methods such as MPEG4 [28], a video is decomposed into appearance and motions (+ residuals). Then, entropy coding is performed to assign short codes to small motions and residuals, which have little information content. Inspired by this, our core idea is to exclude the input patches of P-frames that contain little information. We show the diagram of IPS in Figure 3.

The (self-)information for the *k*-th patch of P-frames created in the manner described in Section 3.2.2 is as follows: (10)$$\begin{array}{ccc} I(x_{\left(k\right)}^{P}) & = & -\log P(x_{\left(k\right)}^{P}) \end{array}$$(11)$$\begin{array}{ccc} & = & -\log P(x_{\left(k\right)}^{M},x_{\left(k\right)}^{R}), \end{array}$$
where $x_{\left(k\right)}^{M}\in R^{P^{2}\times 2}$ and $x_{\left(k\right)}^{R}\in R^{P^{2}\times 1}$ are motion and residual components of $x_{\left(k\right)}^{P}$ reshaped to be decomposed into the spatial and channel dimensions. It should be noted that the residuals are averaged in the channel direction, as shown in Section 3.2.2. Furthermore, we assume that $x_{\left(k\right)}^{M}$ and $x_{\left(k\right)}^{R}$ are independent, and their spatial averages, ${\bar{x}}_{\left(k\right)}^{M}\in R^{2}$ and ${\bar{x}}_{\left(k\right)}^{R}\in R$, are assumed to follow two-dimensional and one-dimensional Gaussian distributions, respectively. We modify the joint distributions in (11): (12)$$\begin{array}{ccc} P(x_{\left(k\right)}^{M},x_{\left(k\right)}^{R}) & = & P\left(x_{\left(k\right)}^{M}\right)\times P\left(x_{\left(k\right)}^{R}\right)\\ & = & N_{2}\left({\bar{x}}_{\left(k\right)}^{M};\mu^{M},\Sigma^{M}\right)\times N_{1}\left({\bar{x}}_{\left(k\right)}^{R};\mu^{R},{\left(\sigma^{R}\right)}^{2}\right), \end{array}$$
where $N_{1}\left(\cdot ;\mu ,\sigma^{2}\right)$ is the one-dimensional Gaussian distribution of the mean μ variance $\sigma^{2}$ and $N_{2}\left(\cdot ;\mu ,\Sigma \right)$ is the two-dimensional normal distribution of the mean μ covariance matrix Σ. We set $\mu^{M}=\left[0,0\right]$ and $\mu^{R}=0$, and $\Sigma^{M}$ and ${\left(\sigma^{R}\right)}^{2}$ are calculated from the training data of the dataset.

Finally, a set of patches whose self-information is above the threshold are selected as input: (13)$$\begin{array}{c} S\in \{k|I\left(x_{\left(k\right)}^{P}\right)>\tau \}, \end{array}$$
where the threshold τ is a hyper-parameter set by users according to their computational constraints. More precisely, users can calculate the distribution of $I(x_{\left(k\right)}^{P})$ with (a part of) training data and determine the threshold that achieves the desired patch reduction ratio based on this distribution. Figure 4 shows histograms of information $I(x_{\left(k\right)}^{P})$ for each patch of 1000 videos extracted from the HMDB-51 train and validation set, respectively. It can be seen that the distributions of the information for the training and validation sets are very close. Moreover, in Table 1, we show the patch reduction ratios in the validation set (1000 videos) by the thresholds, which are determined to achieve specific patch reduction ratios in training set (1000 videos). The discrepancy between the patch reduction ratios in the training set and those in the validation set is minimal, indicating that the desired patch reduction ratio can be achieved with high accuracy, even for videos not included in training set. Note that in our experiments described in Section 4, we adjusted the inference cost by varying the threshold to evaluate the trade-off curve between the accuracy and inference cost.

Figure 5 shows examples of GoPs in HMDB-51 processed by IPS. For static scenes, IPS efficiently selects only the areas with motions or changes. When large scene transitions occur, such as a cut, IPS selects an entire frame so that enough information about the new scene can be input into the model, and thereafter selects only the moving or changing patches for efficiency (see the first GoP in Figure 5).

#### 3.4.1. P-Frame Accumulation

A problem arises when some of the patches represented by motions and residuals are excluded from the input using the method presented above. That is, it is often impossible to identify regions in an I-frame (appearance) that correspond to observed patches in a P-frame due to interruptions in the motion trajectories by unobserved patches. This means that the model may not associate appearances and motions or reconstruct the RGB values of the P-frame, which may lead to inaccurate video recognition. To solve this problem, we adopt the ”P-frame accumulation” proposed in [45]. P-frame accumulation backtraces motion trajectory up to the previous I-frame and sums up the motions and residuals to make new accumulated motions and residuals, which depend only on the previous I-frame. Thereby, without recursively reconstructing P-frames, the RGB values of a P-frame can be reconstructed from the previous I-frame, which is always inputted, and the target P-frame itself. Note that we use P-frame accumulation only for input into the video model. *Motions and residuals for IPS are always non-accumulated ones.*

## 4. Experiments

We evaluate our method and perform comparisons with baseline models on action recognition benchmarks, where the methods are required to take a video as input and output the action class occurring in the video (i.e., classify videos into the action classes). This section describes the experimental setups; the results of an ablation study and comparisons to baselines and state-of-the-art methods; and discussions of these results.

### 4.1. Setups

Here, we describe the datasets, evaluation settings and implementation details of our method, including preprocessing and training procedures. In addition, we present baselines prepared for comparisons to confirm the effectiveness of our method.

#### 4.1.1. Datasets and Evaluation

We used three datasets for action recognition:**HMDB-51 [50]** contains 3570 videos for training and 1530 videos for validation; they are labeled according to 51 action classes. The average duration of the videos is approximately 3.1 s.**Something-Something V2 (SSv2) [51]** includes 168.9k videos for training and 24.7k videos for validation; they are labeled according to 174 classes. The average duration of the videos is approximately 3.8 s.**Mini-Kinetics [33]** is a subset of Kinetics [19], which consists of 200 classes of videos selected from Kinetics, with 121k videos for training and 10k videos for validation. The average duration of the videos is approximately 10 s.

SSv2 is a relatively motion-intensive dataset, whereas HMDB-51 and Mini-Kinetics are more about appearance information. Therefore, we performed a detailed ablation study and comparisons to baselines using HMDB-51 and SSv2, and we conducted comparisons to state-of-the-art methods using HMDB-51, SSv2, and Mini-Kinetics.

We used GFLOPs per video as a device-independent metric of each models’ inference cost . In addition, we measured the runtime of the models’ inference on both a CPU (Central Processing Unit) and GPU, as well as the runtime of video decoding on CPUs. We used an AMD Ryzen 9 3900X 12-Core Processor as the CPU and an NVIDIA GeForce RTX 3090 as the GPU. We mainly compared the models’ inference costs, which does not include the process of decoding compressed videos nor of encoding frame images into compressed videos. We refer to models’ inference costs as the ”inference cost”, unless otherwise specified. The reasons for this are as follows: First, as mentioned in Section 3, we make the realistic assumption that videos are stored in a compressed state; therefore, we can ignore the process of encoding frame images into compressed videos. In addition, based on this assumption, the proposed methods and all of the comparisons in our experiment require decoding a compressed video into RGB values or motions+residuals before the models’ inference, and the difference in the cost of the video decoding processes is relatively smaller than that in the cost of the models’ inference, which will be discussed in Section 4.2.

#### 4.1.2. Model and IPS Implementation

We used the official implementation of TimeSformer [52] pretrained on ImageNet [53,54] as a Transformer-based video model. It should noted that the weights of the linear projections of motions and residuals were initialized randomly. The number of blocks was set to eight to save on computational cost. We call it TimeSformer-S to distinguish it from the original TimeSfomer (12 blocks). P-frame accumulation was applied for input into the TimeSformer-S, unless otherwise specified.

#### 4.1.3. Preprocess and Training

To efficiently take into account the long-term relationship, the entire video in the dataset was first re-encoded at 6 fps. The input was a clip of 24 frames (i.e., 2 GoPs, 4 s) randomly extracted from the entire video for training and at the temporal center of the video for testing. We also applied spatial cropping to obtain a square region. According to [2], during training, as a form of data augmentation, we first randomly resized the shorter side of the frame image to [1.0,1.2) times the final crop size while retaining the aspect ratio, and we then randomly cropped it to the square region with the final crop size. During testing, we resized the shorter side of the frame image to the final crop size while keeping the same aspect ratio and cropped the center of the frame. Unless otherwise specified, we set the final crop size to 144×144.

As a loss function for classification, we used cross entropy. We performed 15 epochs of training using stochastic gradient descent [55] with a momentum of 0.9 for all settings. The initial learning rate was set to 0.005 and the learning rate was divided by 10 at 10 and 14 epochs.

#### 4.1.4. Baselines of Patch Selection

Multiple patch selection baselines were prepared for comparison to confirm the effectiveness of IPS:**Uniform space (uni space):** The spatial indices of patches were sampled uniformly at equal intervals, and the patches with those spatial indices were excluded from the input in all the frames.**Uniform time (uni time):** The time indices of patches were sampled uniformly at equal intervals, and all patches in those time indices were excluded from the input.**Uniform (uni):** The spatiotemporal indices of the patches were randomly selected according to a uniform distribution and excluded from the input. Compared to uni space, the range of spatial indices that can be covered tended to be larger, because different spatial indices were allowed to be selected at different time indices. Similarly, compared to uni time, the range of time indices that could be covered tended to be larger.**Frame subtraction (sub):** Patches were selected if the average absolute subtraction from the previous frame was greater than or equal to a threshold value.

The inference cost was adjusted by varying the sampling rate for uni space, uni time and uni and by varying the threshold for sub.

### 4.2. Effectiveness of IPS for Each Attention Type

First, we demonstrate the effectiveness of IPS in adjusting the inference cost for TimeSformer for each attention type. Figure 6 illustrates that IPS can reduce the inference cost and that its efficiency is very high for all attention types. For example, using Joint Space-Time attention, IPS could achieve a 71.8% reduction in inference cost with a drop in accuracy of only 0.89% on the HMDB-51 dataset and a 69.2% reduction in inference cost with a drop in accuracy of only 1.93% on the SSv2 dataset.

Notably, IPS is a more flexible and efficient way to reduce inference cost than the engineering attention matrix calculations such as Divided Space-Time and Factorized Encoder is in such a low-computation domain (note the comparison of the inverted triangles in the figure).

Second, we show the trade-off between the accuracy and total inference runtime (including the models’ inference and video decoding processes) when using IPS on both CPUs and GPUs for each attention type in Figure 7. Even though GFLOPs is a reliable device-independent measurement of inference cost, it is not necessarily proportional to runtime on specific devices. Figure 7 illustrates that IPS can significantly reduce not only GFLOPs but also runtime from the baselines on both GPUs and GPUs. We guess that the reason for the high minimum runtime of Divided Space-Time attention on a GPU is that its spatial and temporal attentions are applied sequentially and cannot be parallelized. In addition, the large number of tensor reshaping processes due to the use of multiple types of axis-attention is thought to be a bottleneck in both Divided Space-Time attention and Factorized Encoder. Because of the efficiency of the runtime on CPUs and the number of parameters (58M for Joint Space-Time attention, 82M for Divided Space-Time attention and 65M for Factorized Encoder), Joint Space-Time attention was used unless otherwise specified.

In addition, we show a comparison of the costs of video decoding and model inference in Table 2. Decoding P-frames as motions+residuals, which our method requires, takes slightly more runtime than decoding all frames as RGB due to some overhead. However, we can see that the difference in the costs of video decoding is relatively smaller than that in the costs of model inference (especially on CPUs). Even considering this slight increase in the cost of decoding videos, the reduction in the total inference cost by IPS is significant.

### 4.3. Ablation Study

In this section, we explore four design choices for our overall approach, input modalities for the model and the use of P-frame accumulation, residuals and distribution fitting for IPS.

#### 4.3.1. Input Modalities

First, we examine which modality should be used to represent P-frames *without using IPS*. P-frames can be represented as RGB, motions + residuals or motions + residuals with P-frame accumulation. The accuracies in the HMDB-51 and SSv2 datasets using each type of P-frame are shown in Table 3. In both datasets, motions+residuals and motions+residuals with P-frame accumulation had better accuracies than RGB, especially in SSv2, in which motion is more critical. The use of P-frame accumulation caused a slight decrease in accuracy. Next, we investigate the effect of accumulation when IPS is used.

#### 4.3.2. P-Frame Accumulation

Figure 8 (IPS vs. IPS w/o accum) illustrates a comparison of the trade-offs between accuracy and inference cost adjusted via IPS with P-frame accumulation and without it. It is evident that the use of P-frame accumulation exhibited a better trade-off, particularly in a range where the inference cost is low. We believe this is because, as described in Section 3.4.1, when P-frame accumulation is not used, if some patches are excluded, the model cannot associate appearance (i.e., I-frame) with motions. The effect of this problem increases as the number of patches to be excluded increases. P-frame accumulation avoids this problem.

#### 4.3.3. Residuals

Figure 8 (IPS vs. IPS w/o res) compares the trade-offs between accuracy and inference cost adjusted via IPS with residuals and without them. In the absence of residuals, Equation (Equation 12) is only the probability of the motions (i.e., $P(x_{\left(k\right)}^{R})=1$). It is evident that using not only motions but also residuals in IPS results in a good trade-off. If the motion is small but the residual is large, it may be difficult to predict from the previous frame; that is, the information content may be large. On the other hand, if the motion is not small but the residual is small, the patch may be predictable to some extent; that is, the information content may be small. Therefore, it is necessary to consider both motions and residuals to correctly estimate the amount of information in a patch.

#### 4.3.4. Distribution Fitting

Figure 8 (IPS vs. IPS w/o dist. fitting) illustrates a comparison of the trade-offs between accuracy and inference cost adjusted via IPS with fitting variance parameters of Gaussian distribution for motions and residuals ($\Sigma^{M}$ and ${\left(\sigma^{R}\right)}^{2}$) in Equation (Equation 12)) or without it. ”IPS w/o dist. fitting” denotes the case in which the variances of the motions and the residuals are set to ${\left(\sigma^{R}\right)}^{2}=1$, $\Sigma^{M}=I_{2}$ (where $I_{2}$ is the 2×2 identity matrix). Without distribution fitting, a slight performance drop was observed. Although IPS is somewhat robust for this parameter setting, tuning should be performed if possible. In particular, the scales of motions and residuals are different; therefore, it is important to appropriately balance their contribution to the information content calculation with the variance parameters.

### 4.4. Comparison with Baselines

Figure 9 illustrates a comparison of accuracy and inference cost with the baselines designed in Section 4.1.4. Compared to the baselines designed to select patches uniformly (uni space, uni time and uni), IPS exhibited significantly better performance, indicating that it is effective to consider the amount of information in the patches. The baseline sub, which selects patches based on absolute frame subtraction, performed better than uniform selection, but not as well as IPS. Compared to simple frame absolute subtraction, IPS offers a more sophisticated definition of the amount of patch information, considering the nature of videos, in which each frame is composed by adding motions to the previous frame.

Changing the resolution is another method of reducing the input size. Figure 10 shows the results of applying IPS for each input resolution. It is evident that the trade-offs of using IPS were significantly better than the trade-off of changing the resolution (note the curve drawn by the inverted triangle in the figure).

### 4.5. Comparison with State-of-the-Art Methods

Table 4, Table 5 and Table 6 show the comparisons with state-of-the-art methods on the HMDB-51, SSv2 and Mini-Kinetics datasets, respectively. We selected both Transformer-based and CNN-based methods as comparisons. For Transformer-based video models, we select the recent major models, ViViT [3], VidTr [4], VideoSwin [49] and MM-ViT [5], in addition to our base model, TimeSformer [2]. A detailed introduction for these models can be found in Section 2. For CNN-based models, we select major methods that focus on realizing a *better trade-off between accuracy and inference cost in the low-computation domain. These can be categorized into two types of approaches, static to input and dynamic to input.* TSN [14], TRN [15], ECO [47], TSMNet [16,17] and MSNet [48] are static to input and therefore require no policy model other than the classification model. The other CNN-based methods we picked (Adafuse [35], Dynamic-STE [40], FrameExit [39], LiteEval [29], SCSampler [32], ARNet [36], VideoIQ [34] and Adafocus v1-3 [36,37,38]) include dynamic optimization of inferencing. They optimize the input-size or inference pipeline dynamically from each viewpoint and require a policy model (and additional loss functions to train them) to adjust inference other than the classification model. In contrast, our proposed method is dynamic to input yet does not require a policy model, making it easier to use. We also introduced the CNN-based methods described above in Section 2.

First, among the Transformer-based video models, which tend to be expensive, TimeSforer-S with IPS achieved a very lightweight inference cost (Table 5). Although it is slightly less accurate than the conventional Transformer-based video model, the accuracy difference from the original TimeSformer (”TimeSformer” in Table 5) was minimal. In addition, we believe that reducing the number of patches via IPS is also effective for reducing the inference cost of other methods, which model spatiotemporal patches of a video with Transformers, such as ViViT [3], VidTr [4] and MM-ViT [5].

On the HMDB-51 dataset, which contains only a relatively tiny amount of data, the accuracy of TimeSformer-S is below the level of accuracy of CNN-based models (Table 4). This is due to the well-known fact [7,22] that Transformer-based models require more training data than CNNs to fully realize their potential since they lack inductive bias in exchange for strong expressivity In contrast, on the SSv2 and Mini-Kinetics datasets, which are relatively large-scale, IPS achieved an efficiency comparable to those achieved by CNN-based models specializing in low computational cost (Table 5 and Table 6). Some CNN-based dynamic to input methods, especially Adafocus v2 and v3, achieve strong trade-offs on both SSv2 and Mini-Kinetics. However, they require additional policy models, making their training procedures somewhat complicated. We would like to emphasize that IPS—which can be considered a simple preprocessing method that requires no policy models or additional losses—has an obvious benefit over those CNN-based dynamic to input methods in terms of simplicity and ease of use. Despite the simplicity of TSMNet and MSNet, which require no policy model, they are highly efficient in SSv2. These are the result of the many improvements to ensure that the architecture is compatible with video data. Since Transformer has higher expressivity than CNN, it has the potential to scale up in accuracy under conditions with more training data. With the development of data collection pipelines from the Internet, video datasets for training/pre-training models will be thought to continue to grow in size, and Transformer-based video models will be able to enjoy them. We believe that IPS is essential to such anticipated video models.

## 5. Conclusions

We propose IPS, a method to efficiently reduce the inference cost by excluding redundant patches from the input of a Transformer-based video recognition model. Inspired by video compression, we define the redundancy of a patch using motions and residuals, which we obtain while decoding a compressed video. Extensive experiments on action recognition demonstrated that our method could significantly improve the trade-off between the accuracy and inference cost of a Transformer-based video model. Despite its simplicity, the efficiency of the model equipped with IPS approaches that of more complicated existing methods requiring policy models or additional losses.

This research is a pioneering attempt to develop a method that takes a compressed video as input and adjusts the inference cost based on the amount of its information. We believe that our idea and experimental results will open up new possibilities for improving the efficiency of Transformer-based video models, which have the weakness of huge inference costs. Even though IPS assumes Transformer-based video models, the potential of which is not fully realized with the limited training data, it is expected that video datasets for training/pre-training will continue to grow in size, allowing Transformer-based video models to scale up in accuracy. One limitation of the proposed method is that currently it only supports the MPEG4 video codec. Even though MPEG4 is a widely used video codec in the real world, further support for other codecs will allow the proposed concept to have a wider range of applications. Specifically, we would like to support H.264 [57] and HEVC [58]), as well as neural-network-based video compression [59], which we will aim for in future work. By utilizing these more efficient compression methods, we expect to improve the efficiency of video recognition using IPS. Furthermore, developing video representations optimal for both compression and recognition is to be considered for future research.

## Figures and Tables

**Figure 1 sensors-23-00244-f001:**
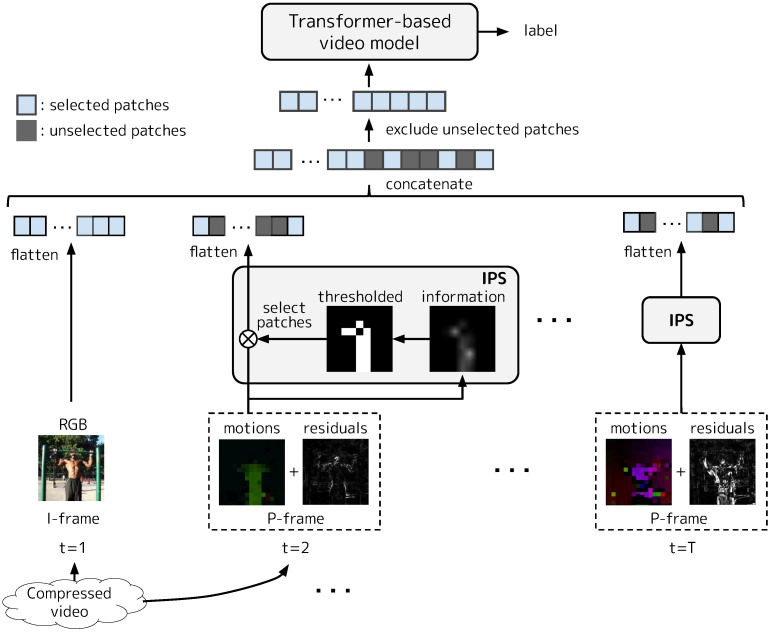
Overview of Transformer-based compressed video modeling with IPS. First, I-frames and P-frames in a compressed video are decoded into RGB values and motions+residuals, respectively. Second, both the I-frames and the P-frames are divided into patches. Then, while all the patches are inputted into a Transformer-based video model for the I-frames, for P-frames, only patches selected based on their information content by IPS (see Section 3.4 for details) are inputted. Finally, the Transformer-based model outputs some label (e.g., action class) associated with the input video.

**Figure 2 sensors-23-00244-f002:**
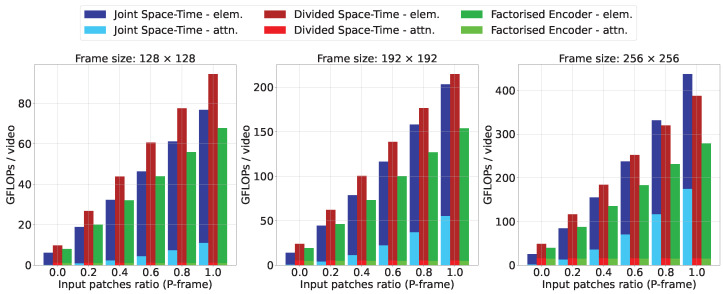
Computational cost of attention matrix calculation (”attn”) and element-wise calculation (”elem”) in TimeSformer for each resolution and type of attention when changing the percentage of reduction in the number of patches for P-frames. The number of blocks in the Transformer is set to 12, the input is 1 GoP (i.e., 1 I-frame and 11 P-frames), and the patch size is uniformly set to 16.

**Figure 3 sensors-23-00244-f003:**
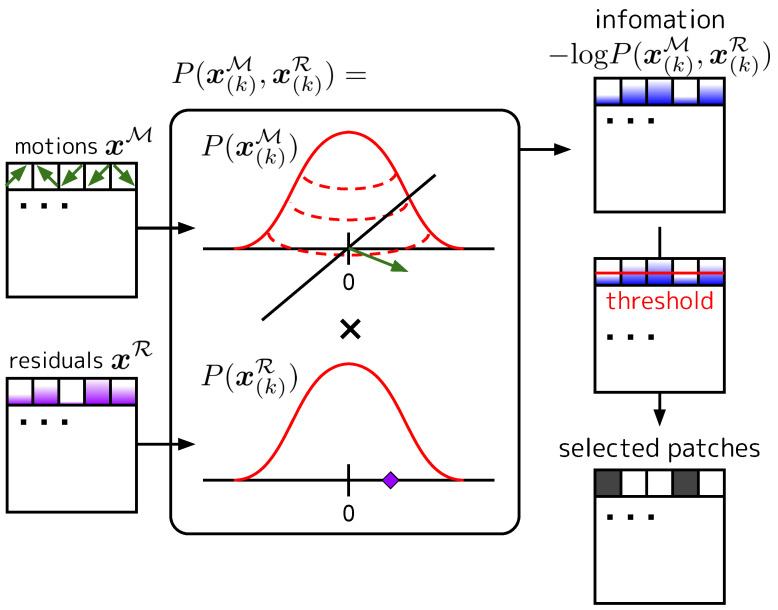
Diagram of IPS. We model the distributions of the motions and residuals in a patch of P-frames and calculate the amount of information content in each patch. Patches with information content above the threshold are selected as input into the video model.

**Figure 4 sensors-23-00244-f004:**
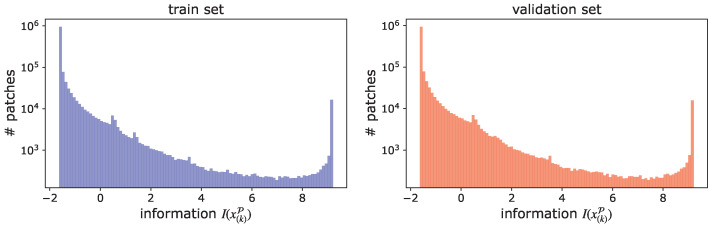
Histograms of information $I(x_{\left(k\right)}^{P})$ for each patch of 1000 videos extracted from the HMDB-51 training and validation sets. The distributions of information $I(x_{\left(k\right)}^{P})$ for the training and validation sets are very close.

**Figure 5 sensors-23-00244-f005:**
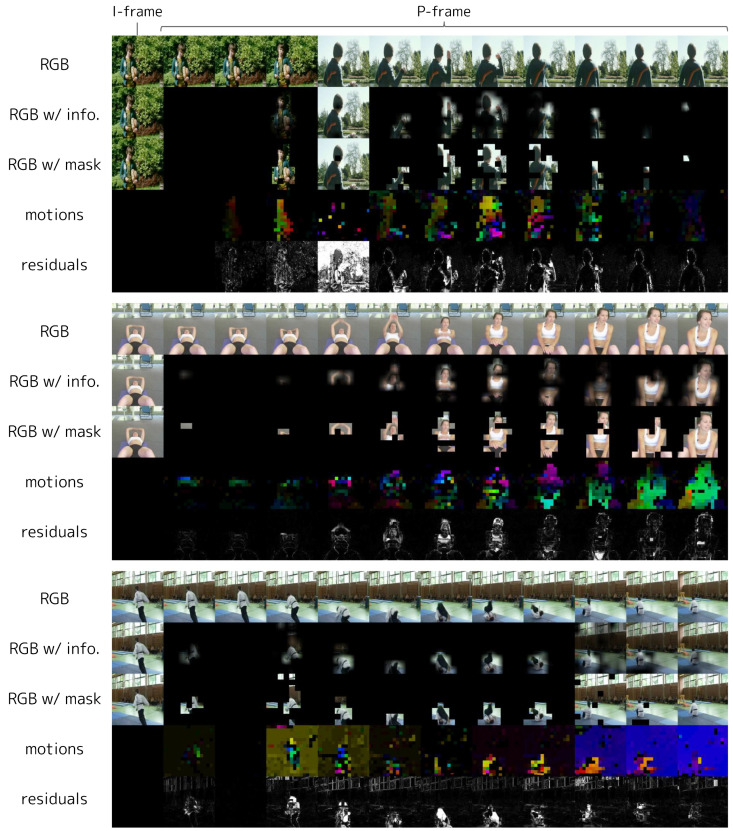
Visualization of GoPs processed by IPS in HMDB-51. Each GoP is displayed by a group of five rows (three GoPs are displayed). The first row of each GoP represents the original frames, the second represents the frames overlaid with information content calculated by Equation (Equation 10), the third represents the frames overlaid with the mask obtained by thresholding the information content, the fourth represents the motions, and the fifth represents the residuals. We plot motions in HSV space, where the H channel encodes the direction of motion, and the S channel shows the amplitude. For residuals, we plot the mean absolute values of RGB. All rows are in chronological order from left to right. The leftmost frames are I-frames and, therefore, not subject to IPS.

**Figure 6 sensors-23-00244-f006:**
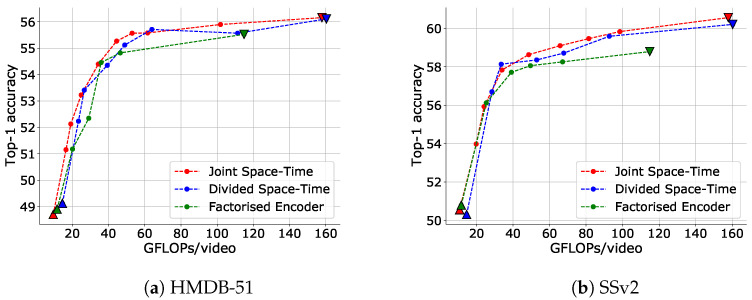
Trade-off between top-1 accuracy and inference cost adjusted via IPS for each attention type of TimeSformer. An input clip has 24 frames with a frame size of 144×144 and 6 fps. The data point with the highest FLOPs for each setting is when no input reduction is performed (upper bound), and the data point with the lowest FLOPs is when only I-frames are input (lower bound). The upper and lower bounds are plotted as inverse triangles and triangles, respectively.

**Figure 7 sensors-23-00244-f007:**
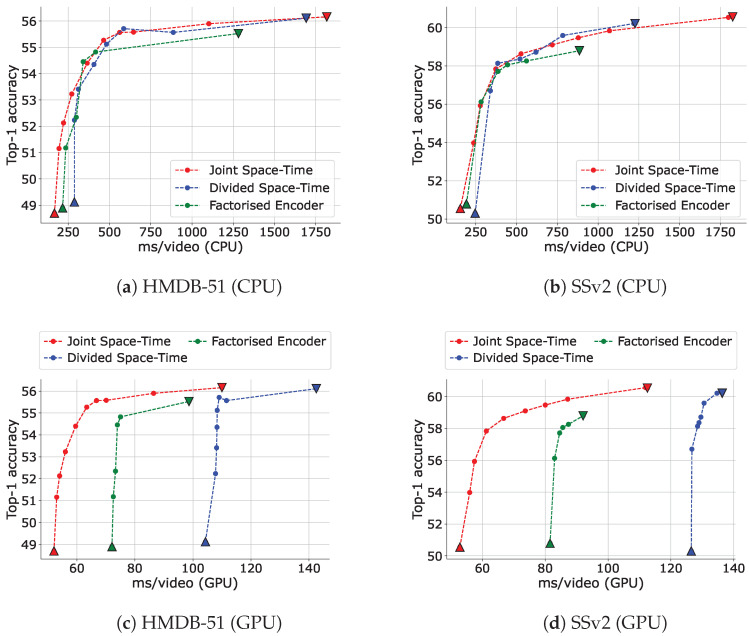
Trade-off between top-1 accuracy and CPU and GPU runtime adjusted via IPS for each attention type of TimeSformer. The runtime includes those of the model’s inference and video decoding, and video decoding was always run on a CPU. ”ms” stands for milliseconds. We used AMD Ryzen 9 3900X 12-Core Processor as CPU and NVIDIA GeForce RTX 3090 as GPU. An input clip has 24 frames with a frame size of 144×144 and 6 fps. The data point with the highest runtime for each setting is when no input reduction is performed (upper bound), and the data point with the lowest runtime is when only I-frames are input (lower bound). The upper and lower bounds are plotted as inverse triangles and triangles, respectively.

**Figure 8 sensors-23-00244-f008:**
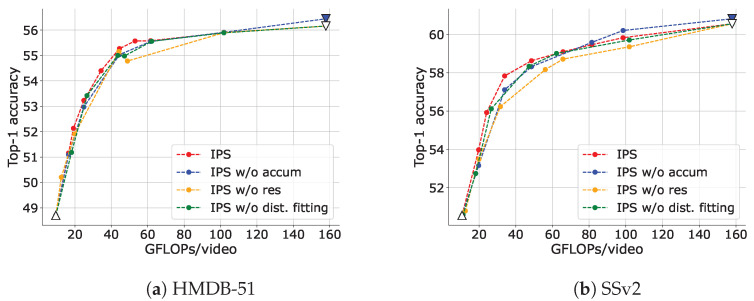
Trade-offs between top-1 accuracy and inference cost adjusted via IPS and IPS without each element. Here, ”accum” stands for P-frame accumulation, ”res” for residuals and ”dist. fitting” for distribution fitting. The data point with the highest FLOPs for each setting is obtained when no input reduction is performed (upper bound), and the data point with the lowest FLOPs is obtained when only I-frames are input (lower bound). The upper and lower bounds are plotted as inverse triangles and triangles, respectively. In these comparisons, the upper bound is the same for ”IPS”, ”IPS w/o res” and ”IPS w/o dist. fitting”, and the lower bound is the same for all settings. They are plotted as a white-filled inverted triangle and a triangle, respectively.

**Figure 9 sensors-23-00244-f009:**
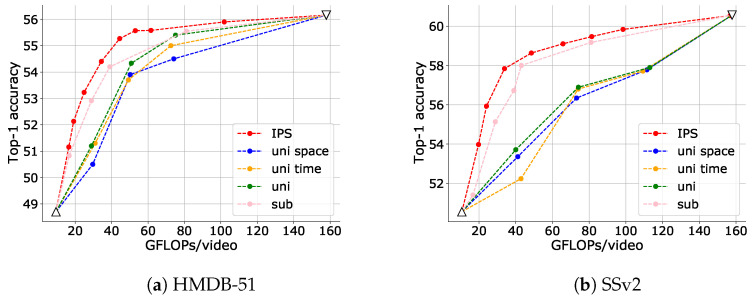
Comparisons of trade-offs between top-1 accuracy and inference cost adjusted via IPS and the baselines. The data point with the highest FLOPs for each setting is obtained when no input reduction is performed (upper bound), and the data point with the lowest FLOPs is obtained when only I-frames are input (lower bound). In these comparisons, the upper and lower bounds are the same for all settings, and they are plotted as a white-filled inverted triangle and a triangle, respectively.

**Figure 10 sensors-23-00244-f010:**
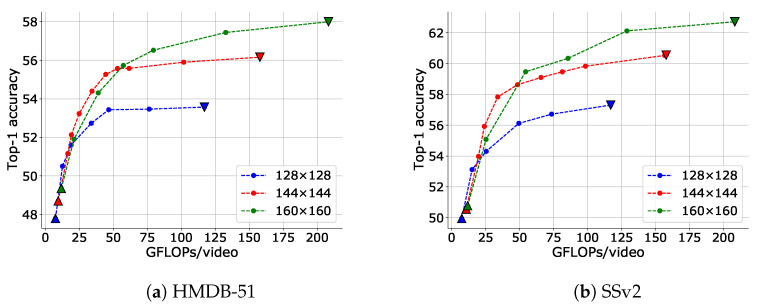
Trade-offs between top-1 accuracy and inference cost adjusted via IPS for different input frame resolutions. The data point with the highest FLOPs for each setting is obtained when no input reduction is performed (upper bound), and the data point with the lowest FLOPs is obtained when only I-frames are input (lower bound). The upper and lower bounds are plotted as inverse triangles and triangles, respectively.

**Table 1 sensors-23-00244-t001:** The patch reduction ratios in the validation set (1000 videos) by the thresholds, which are determined to achieve specific patch reduction ratios in the training set (1000 videos) on HMDB-51.

Split	Patch Reduction Ratios		
training set	30%	60%	90%
validation set	29.85%	59.35%	89.58%

**Table 2 sensors-23-00244-t002:** Comparison of costs per video for decoding and model inference on HMDB-51. The recognition model is TimeSformer-S with Joint Space-Time attentions. ”ms” stands for milliseconds. We used AMD Ryzen 9 3900X 12-Core Processor as CPU and NVIDIA GeForce RTX 3090 as GPU. An input clip has 24 frames with a frame size of 144×144 and 6 fps.

Process	CPU Runtime (ms)	GPU Runtime (ms)	GFLOPs
video decoding (P-frame as RGB)	32.230	–	–
video decoding (P-frame as motions + residuals)	39.046	–	–
model inference	1775.296	70.803	158
model inference with IPS	153.209	18.778	30

**Table 3 sensors-23-00244-t003:** Top 1 accuracies on HMDB-51 and SSv2 using each type of P-frame for input into model. Here, we do not use IPS. The abbreviation ”accum” indicates P-frame accumulation.

P-Frame as	HMDB-51	SSv2
RGB	55.11	56.10
motions + residuals	56.45	59.92
motions + residuals w/accum	56.16	59.56

**Table 4 sensors-23-00244-t004:** Comparison with state-of-the-art methods on HMDB-51. ”IN” and ”K400” represent ImageNet [53,54] and Kinetics-400 [19], respectively. Our methods appear in boldface. ”mo+res” stands for taking P-frames as motions and residuals. All of our methods take 24 frames with the frame size of 144×144 and 6fps.

Base ops.	Methods	Pretrain	HMDB-51	
			GFLOPs	Top-1 acc
convolution	TSN [14]	K400	33	65.1
	ECO [47]	K400	64	72.4
	TSMNet [16,17]	K400	33	73.2
	MSNet [48]	IN+K400	34	75.8
Transformer	**TimeSformer-S**	IN	158	55.1
	**TimeSformer-S (mo+res)**	IN	158	56.2
	**TimeSformer-S (mo+res, IPS)**	IN	34	54.4

**Table 5 sensors-23-00244-t005:** Comparison with state-of-the-art methods on SSv2. ”IN” and ”IN21k” represent ImageNet and ImageNet-21k [53,54], respectively. ”K400” denotes Kinetics-400 [19]. Our methods appear in boldface. ”mo+res” stands for taking P-frames as motions and residuals. All of our methods take 24 frames with the frame size of 144×144 and 6fps.

Base ops.	Methods	Pretrain	SSv2	
			GFLOPs	Top-1 acc
convolution	TSN [14]	K400	33	30.0
	TRN [15]	IN	32	55.5
	TSMNet [16,17]	K400	33	59.1
	MSNet [48]	IN+K400	34	63.0
	Adafuse [35]	IN	32	59.8
	Adafocus v1 [36]	IN	34	60.7
	Adafocus v2 [37]	IN	34	61.3
	Adafocus v3 [38]	IN	15	59.6
Transformer	TimeSformer [2]	IN	590	59.5
	TimeSformer-HR [2]	IN	5110	62.2
	TimeSformer-L [2]	IN	7140	62.4
	ViViT-L [3]	IN21k	5800	65.4
	VidTr-M [4]	IN21k	179	61.9
	MM-ViT [5]	IN21k	2250	64.9
	VideoSwin-B [49]	K400	321	69.6
Transformer	**TimeSformer-S**	IN	158	57.1
	**TimeSformer-S (mo+res)**	IN	158	60.6
	**TimeSformer-S (mo+res, IPS)**	IN	30	57.8

**Table 6 sensors-23-00244-t006:** Comparison with state-of-the-art methods on Mini-Kinetics. ”IN” and ”AS” represent ImageNet [53,54] and Audioset [56], respectively. Our methods appear in boldface. ”mo+res” stands for taking P-frames as motions and residuals. All of our methods take 24 frames with the frame size of 144×144 and 6fps.

Base ops.	Methods	Pretrain	Mini-Kinetics	
			GFLOPs	Top-1 acc
convolution	Adafuse [35]	IN	23	72.3
	Dynamic-STE [40]	IN	18	72.7
	FrameExit [39]	IN	20	72.8
	LiteEval [29]	IN	99	61.0
	SCSampler [32]	AS+IN	42	70.8
	ARNet [33]	IN	32	71.7
	VideoIQ [34]	IN	20	72.3
	Adafocus v1 [36]	IN	27	72.2
	Adafocus v2 [37]	IN	27	75.4
	Adafocus v3 [38]	IN	18	75.0
Transformer	**TimeSformer-S**	IN	158	72.9
	**TimeSformer-S (mo+res)**	IN	158	73.1
	**TimeSformer-S (mo+res, IPS)**	IN	30	71.9

## Data Availability

Data are available in a publicly accessible repository. The data presented in this study are openly available from reference number [35,50,51].

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
