# Peer review of "Efficient Transformer-Based Compressed Video Modeling via Informative Patch Selection"

_sensors, 2022, doi:10.3390/s23010244_

Round 1

Reviewer 1 Report

Introducing “Efficient Transformer-based Compressed Video Modeling via Informative Patch Selection” the author(s) did a good work. However, the following points are requested to promote the manuscript’s outcome.

1.      Abbreviation described once at its first appear then used whole over the manuscript (i.e.IPS, CNN, P-frame, ..etc).

2.      Append a paragraph at the end of Introduction to characterized how the rest of the paper organized?

3.      Colorized and enrich the references (34 out of 42 are of type in Proceedings…) used.

4.      Clarify your method’s main drawback?

5.      Evaluating such systems require measured modelling time and calculating the error rates factor(s)? Explain?

I can accept the work after amendment,

Author Response

First of all, we thank the reviewers for their constructive comments. We did our best to revise the manuscript, including the parts they pointed out. In our revised manuscript, we highlight the added parts in red and those removed in blue with a strike-out line. Please see the attached cover letter for the details of our responses to the comments. 

Please note that 

  • "L" denotes "line" in the cover letter. For example, L10-20 means from line 10 to line 20.
  • the cover letter's indices of lines, Figures, Tables, and Equations are based on the revised manuscript, not the first submitted version.

Reviewer 2 Report

This manuscript proposes an Informative Patch Selection (IPS) method that models the distribution of the motions and residuals in a patch of predicted frame and reduces the inference cost of Transformer-based video models by excluding patches with little information content from the input. The visualizations help to understand the methodology, and the ablation experiments are sufficient. However, in my opinion, the authors are required to address the following issues before a possible publication.

Some major concerns:

1.    This method needs to compress the video into motion and residual information in advance. Is the running cost of this part considered when calculating the inference cost?

2.    How to set the threshold for selecting blocks? What is the range of threshold? Are there experiments that analyze and illustrate this.

3.    As can be seen in Figures 3 and 4, the results of the proposed method in this manuscript are inferior to convolution-based methods in terms of inference cost and accuracy. Therefore, the efficiency claimed by the author is only compared with the Transformer-based method. So, what are the advantages of the proposed method compared to convolution-based methods?

4.    It is recommended that Sections 3 and 4 be combined. The description of Figure 1 does not introduce the content of the figure very well, and it is recommended to revise the description of Figure 1.

5.    This manuscript only presents results comparing state-of-the-art methods on datasets SSv2 and Mini-Kinetics. Comparisons with state-of-the-art methods on the HMDB-51 dataset are missing.

6.    Some relevant methods are missing, such as “TCSPANet: Two-Staged Contrastive Learning and Sub-Patch Attention Based Network for PolSAR Image Classification”.

Minor concerns:

1.    Note the English usages in Figure 3, such as “firstly” and “then”.

2.    Note the use of articles, such as lines 281 and 282.

3.    “acc” generally refers to accuracy (such as in Tables 2 and 3), and the author indicates P-frame accumulation (Table 1). Changes to non-misleading abbreviations are recommended.

Author Response

(The authors gave the same response as above.)

Reviewer 3 Report

This study proposes a new method, namely IPS, that reduces the inference cost by excluding redundant patches from the input of the Transformer-based video model. Sufficient background knowledge and detailed comparisons have been provided. In general, I reckon this manuscript’s outcomes are convincing. However, some minor issues need to be addressed and fixed.

-          Some content in the Introduction that details the proposed method is supposed to be in the Methodology part.

-          There are too many subsubsections in the Methodology and Experiments. I suggest the authors give a short summary before those making it easy to understand and locate relevant info. For example, before the comparisons, explain the process etc., to offer an overview, then give details in those subsubsections.

-          In Table 3, the dataset is Mini-Kinetics, but the authors claimed that they “performed a detailed evaluation of our method using HMDB-51 and SSv2”, did I miss something here?

-          The authors have provided very detailed comparisons, but I’d like more details about the selection criteria of those methods. There are no even explanations of those methods selected in the present format.

Author Response

(The authors gave the same response as above.)

Round 2

Reviewer 2 Report

Thanks to the authors for their responses. In summary, the authors' revised manuscript and responses address most of my concerns. However, I still have some questions before it can be accepted:

1.    In Figure 1, what do black and white represent in the features after IPS and flatten? From Figure 1, I don't see that the number of input patches of Transformer-based video model is reduced. Is there some detail missing in Figure 1? Why does the Transformer-based video model become efficient if the number of input patches does not decrease? If the number of input patches is reduced, why is it not shown in Figure 1?

2.    There are too few comparative methods in Table 3, and the performance of the author's method is very low, although the author only Pretrain on IN. Can the author appropriately add the experimental results with K400 as Pretrain?

3.    There is no IN21K in Table 3 and Table 5, but it is introduced in the title of Table 3 and Table 5.

Author Response

(The authors gave the same response as above.)
